Changes in species diversity of arboreal spiders in Mexican coffee agroecosystems: untangling the web of local and landscape influences driving diversity

Hajian-Forooshani Zachary 1 zhajianf@umich.edu
Gonthier David J. 2
Marín Linda 2
Iverson Aaron L. 1
Perfecto Ivette 2
1 Department of Ecology and Evolutionary Biology, University of Michigan , Ann Arbor, MI , United States
2 Department of Natural Resources and Environment, University of Michigan , Ann Arbor, MI , United States
Higley Leon
Electronic publication date: 2014 Nov 4
Publication date: 2014
Volume: 2
Electronic Location ID: e623
Received 2014 Feb 6; Accepted 2014 Sep 25
Copyright: © 2014 Hajian-Forooshani et al.
Copyright year: 2014
Copyright holder: Hajian-Forooshani et al.
License: This is an open access article distributed under the terms of the Creative Commons Attribution License, which permits unrestricted use, distribution, and reproduction in any medium, provided the original author and source are credited.
License URL: https://creativecommons.org/licenses/by/3.0/

Keywords: Agroecosystem, Coffee, Arboreal spiders, Biodiversity, Shade trees, Management, Climate change, Agriculture

Funding: National Science Foundation DEB-1262086 R000713 Travel and equipment funds for Zachary Hajian-Forooshani came from the National Science Foundation ED-QUEST Research Experience for Undergraduate program at the University of Michigan. The funders had no role in study design, data collection and analysis, decision to publish, or preparation of the manuscript.

==============================
Agricultural intensification is implicated as a major driver of global biodiversity loss. Local management and landscape scale factors both influence biodiversity in agricultural systems, but there are relatively few studies to date looking at how local and landscape scales influence biodiversity in tropical agroecosystems. Understanding what drives the diversity of groups of organisms such as spiders is important from a pragmatic point of view because of the important biocontrol services they offer to agriculture. Spiders in coffee are somewhat enigmatic because of their positive or lack of response to agricultural intensification. In this study, we provide the first analysis, to our knowledge, of the arboreal spiders in the shade trees of coffee plantations. In the Soconusco region of Chiapas, Mexico we sampled across 38 sites on 9 coffee plantations. Tree and canopy connectedness were found to positively influence overall arboreal spider richness and abundance. We found that different functional groups of spiders are responding to different local and landscape factors, but overall elevation was most important variable influencing arboreal spider diversity. Our study has practical management applications that suggest having shade grown coffee offers more suitable habitat for arboreal spiders due to a variety of the characteristics of the shade trees. Our results which show consistently more diverse arboreal spider communities in lower elevations are important in light of looming global climate change. As the range of suitable elevations for coffee cultivation shrinks promoting arboreal spider diversity will be important in sustaining the viability of coffee.

Introduction

Agriculture has the potential to play a pivotal role in the conservation of biodiversity worldwide, and with around 40% of the terrestrial Earth currently in agricultural land-use (Foley et al., 2005); the need for more effective management of agroecosystems which considers both food production and biodiversity conservation is evident. With growing concerns about the adverse effects of modern agriculture (Foley et al., 2005; Swift et al., 1996; Rockström et al., 2009; Power, 2010), making agroecosystems more habitable to biodiversity will simultaneously address the global decline in biodiversity while maintaining sustainable agricultural production.

Biodiversity in agroecosystems responds to local management factors which include crop density, crop diversity, crop rotations, and chemical inputs (Tscharntke et al., 2005; Batáry, Matthiesen & Tscharntke, 2010). Biodiversity also responds to landscape scale factors such as distance to forest, management of edge habitat, and landscape heterogeneity (Tscharntke et al., 2005; Schmidt & Tscharntke, 2005). Most species respond to some scale of management intensity (Tscharntke et al., 2005; Drapela et al., 2008; Batáry et al., 2008) therefore management at local and landscape scales can have varying impacts depending on the species.

Coffee agroecosystems in the tropics, when traditionally managed with high numbers of shade trees, tend to harbor more biodiversity then intensive coffee agroecosystems (Perfecto et al., 1996). The shade trees in coffee agroecosystems, which are predominantly Inga spp., provide an important source of habitat for arboreal arthropods such as ants and spiders (Perfecto et al., 1996). Furthermore, shade trees in agroecosystems are important in maintaining microclimatic characteristics such as temperature (Jaramillo et al., 2013), which may be important in determining the distribution of both pests and/or biocontrol agents within an agroecosystem. Intensification of coffee often consists of an increase in chemical inputs and a reduction in shade tree diversity, shade tree density, and thus overall canopy complexity. Recent studies in agroforestry systems, such as coffee and cacao, show that increased biodiversity often provides greater biological control of insect pests and diseases (De Beenhouwer, Aerts & Honnay, 2013). In cacao plantations, for example, management for high shade tree density can lead to an increase in the abundance of important generalist predators in cacao trees, such as web-building spiders (Stenchly et al., 2011).

Spiders are generalist predators that can offer important biocontrol services in agriculture (Riechert & Lockley, 1984; Riechert & Bishop, 1990; Riechert & Lawrence, 1997; Symondson, Sunderland & Greenstone, 2002). Spiders prevent and suppress pest outbreaks in arable crops (Riechert & Lockley, 1984; Symondson, Sunderland & Greenstone, 2002), and can persist even when pest numbers are low by feeding on alternative prey items within the agroecosystem (Settle et al., 1996; Symondson, Sunderland & Greenstone, 2002). In some cases diverse assemblages of spiders provide greater pest suppression than simple assemblages (Riechert & Lockley, 1984; Riechert & Bishop, 1990; Riechert & Lawrence, 1997; Symondson, Sunderland & Greenstone, 2002). Given the importance of spiders in providing biocontrol in agriculture, understanding what factors drive spider abundance and richness is critical in understanding how biodiversity can provide valuable ecosystem services in managed landscapes.

Surprisingly in coffee, spiders show an inconsistent response to intensification and typically tend to increase with increased intensification of the agroecosystem. For example, coffee-dwelling spiders are more diverse in intensified coffee agroecosystems (Pinkus Rendón et al., 2006, L Marín & I Perfecto, pers. comm., 2013) , and spiders that live on tree trunks of shade trees have no relationship with canopy cover and distance to forest, while only being affected by tree trunk characteristics (L Marín, pers. comm., 2013). Pinkus Rendón et al. (2006) found spider diversity in the coffee was negatively correlated with tree cover and plant diversity but only in the rainy season. Similarly, in cacao agroforestry systems, Stenchly, Clough & Tscharntke (2012) reported no impact of shade tree density on overall spider species richness or spider abundance, but a positive impact on the abundance of cacao tree canopy inhabiting web-building spiders (Stenchly et al., 2011). Spider’s lack of response or positive response to management intensification contrasts with the response of other taxonomic groups and responses of spiders to intensification of temperate cropping systems (Clough et al., 2005; Schmidt & Tscharntke, 2005). For instance, studies in arable crops show that heterogeneous landscapes and low intensity agricultural practices have a positive effect on spiders (Clough et al., 2005; Schmidt & Tscharntke, 2005; Schmidt et al., 2008). Spiders in tropical agroforestry systems and temperate arable crops appear to respond to different factors, so understanding what makes these assemblages respond differently is important. Furthermore, to our knowledge, all the studies of spiders in coffee agroecosystems to date do not include the arboreal spider assemblages, in particular the spiders inhabiting the shade tree canopies.

To better understand what factors drive arboreal spider diversity, in this study, we investigated how arboreal spiders respond to a spectrum of local management and landscape characteristics at three spatial scales in coffee agroecosystems. We hypothesized that there would be important drivers of spider diversity at all three spatial scales which included tree characteristics (local scale), plot management (broader local scale), and landscape features. We hypothesized that a higher canopy connectedness as a result of high shade tree density leads to an increase in spider species richness and abundance on tree scale due to facilitating habitat access for arboreal spiders. We also hypothesized that a more dense shade tree layer promotes high spider abundance on plot scales, as this positive relationship could be already reported for tropical spiders in cacao agroforestry systems (Korinus, 2007). At the landscape scale we predicted there would be no effect of distance to forest, which has been reported by L Marín & I Perfecto (2013, unpublished data) for leaf litter spiders in coffee and by Stenchly et al. (2011) for arboreal spiders in cacao plantations. However we hypothesized that the degree of forest cover in a 1,000 m radius will have a negative impact on spider richness and abundance based on the assumption that spiders will remain within forest patches and not move to the coffee patches because unmanaged land tend to harbor highly diverse communities (Batáry et al., 2011). Furthermore, at landscape scale elevation has been an effective predictor of spider communities in the tropics (Russell-Smith & Stork, 1994; Stenchly et al., 2011), and we assumed a decrease in abundance and species richness with increasing elevation.

Methods

We conducted our study in the Soconusco region of Chiapas, Mexico across coffee plantations that ranged in elevation between 595 and 1273 m.a.s.l. The Soconusco landscape is dominated by coffee agriculture (94% land cover), with small forest fragments (6% land cover) lining some valleys and mountain ridges (Philpott et al., 2008). We located 38 sites within 9 coffee plantations that varied by management intensity within this region.

Within each site we measured tree scale, site scale, and landscape scale factors (Table 1). Tree scale factors included: tree height (estimated), circumference at breast height (CBH) (measured), branch length (measured), branch diameter at three spots on the branch (measured), number of leaves (counted), and the number of tree canopies touching the sampled tree (counted) and identity of those trees. The average diameter of the branch was estimated and used with branch length to calculate branch volume, which was used as a measure of sampling effort. The leaf area of all leaves per branch was used to estimate total leaf area per branch. Local site scale factors described site characteristics as they pertain to the intensity of the management of the coffee plantation; in particular percent shade cover and shade tree density. We used a Global Positioning System to map a hectare circular area around the center of a site, and then documented the abundance and richness of all tree species within that area. A densiometer was used to measured canopy cover at the center, 5 m and 10 m away in each cardinal direction and used the average of these measurements. Percentage canopy cover was used as a measure of shade on a plot.

Table 1 Mean, minimum, and maximum values for tree level factors, plot level management factors, and landscape level factors.

	min	max	mean	
Tree scale				
Height	400 cm	1,700 cm	743 cm	
CBH	20 cm	206 cm	74 cm	
# of connections	0	6	1.3	
Branch volume	40 cm3	460 cm3	190 cm3	
Leaf area	9,692 cm3	73,549 cm3	31,164 cm3	
Plot scale				
Shade cover	2.50%	94%	94%	
Plot area	0.8 ha	1.1 ha	1.0 ha	
Total trees	57	312	169	
Trees per area	63	337	169	
Landscape Scale				
Forest in 1,000 m	0	18%	7%	
Low intensity agriculture in 1,000 m	0	90%	30%	
Med intensity agriculture in 1,000 m	0	90%	40%	
High intensity agriculture in 1,000 m	0	90%	30%	
Distance to forest	60 m	870 m	321 m	
Landscape heterogeneity in 1,000 m	0.17	15	1	
Elevation	595 m	1273 m	942 m	

Using Geographic Information Systems (GIS) we measured landscape scale factors surrounding each site. To measure landscape composition, we digitized forests and coffee farms of varying intensity using ArcGIS 10 and utilizing a basemap of the region. Plantation boundaries were used to define rough categorizations of landscape management intensity based on the average percent shade cover of plantations: low—(>70%), medium—(30%–70%), and high—(<30%) management. Some plantations had large areas of more than one category of shade intensity scale. We therefore delineated these areas and categorized each area into its appropriate scale. With this categorization, we calculated percent forest, low-shade, medium-shade, and high-shade coffee land-use types within 100, 250, 500, and 1,000 m radii surrounding each site. We also calculated the Shannon diversity index (Σ-ln(p) p) of the habitat types. Although correlations of dependant variables and landscape factors at these four difference scales were similar, we analyzed our data utilizing 1,000 m scale (the largest scale) because we were interested in capturing the greatest difference in local versus landscape scales.

At each site three shade trees were selected belonging to the species Inga micheliana or I. rodrigueziana, the two most common shade species in the region. On five of the sampled sites, we only found two trees of these species therefore we have a total of 109 sampled trees (instead of 114). Once trees were selected, we cut two branches from each tree with an extendable pole-cutter or the tree was scaled and branches lowered down. The branches were then shaken aggressively over a 1 by 1 m black blanket, so that spiders could be more efficiently collected. After shaking no longer produced more spiders, we put the branch on the blanket and all of the leaves were checked for spiders. All spiders were stored in vials of 97% alcohol in the field.

The specimens were sorted into morphospecies and identified to species when possible. For all reproductively mature spiders, body length was measured under a dissection scope. Guillermo Ibarra Nuñez assisted the identification of spiders at El Colegio de la Frontera Sur in Tapachula, Chiapas. Spiders were broken into 5 guilds defined by Young & Edwards (1990), which included sheet-web, orb-web, matrix-web, active-wandering, and ambush-wandering. We condensed these guilds into two groups: web-building spiders and wandering spiders. To determine if our sampling of the spiders was representative of the overall community, we constructed a sample-based rarefaction curve (MaoTao estimations in EstimateS) of spider richness per individual sampled. An asymptotic rarefaction curve suggested our sampling was representative of the community and any additional sampling would yield few new species.

To determine which factors were strong predictors of arboreal spiders, we deployed Conditional inference trees (CIT) and generalized linear mixed models (GLMM). We chose to use the two different statistical methods for two reasons, GLMMs are traditional multi-variate analysis technique within the literature, but it is difficult to incorporte interactions between many independent variables without over-fitting the model. CITs are considered well suited to deal with complex non-linear and high-order interactions in ecological data (De’ath & Fabricius, 2000). CITs are non-parametric tests that are also robust to problems associated with collinearity between independent variables (Piramuthu, 2008).

To determine which factors were strong predictors of arboreal spiders, we first employed GLMMs with Poisson distributions. We used the variables in Table 1 to uncover strong predictors of overall arboreal spider abundance and richness, web-building spider abundance and richness, along with wandering spider abundance and richness. To avoid pseudo-replication, we used site as a random effect in the model. We also explored the option of using farm as a random effect within models because of discrepancy in the number of sites between farms. However this factor did not improve model fit, therefore we proceeded without it. Akiake Information Criterion (AIC) elimination was used to build best-fit models. If a model was less than 2 AIC values from a nested model, the most parsimonious model was chosen. The “lme4” package was used to run all generalized linear mixed models.

We also deployed CITs to determine which factors were strong predictors of arboreal spiders. CITs use a binary recursive data-partitioning algorithm to estimate regression relationships, and do not assume linearities in the response variables. Parameter instability tests are used for split selection in the tree building process (Hothorn et al., 2010). CITs were run with the ‘party’ package in R, which gives p-values at each node of the tree. Six total trees were run: overall arboreal spider abundance, overall arboreal spider richness, web-building spider abundance, web-building spider richness, wandering spider abundance, and wandering spider richness. Independent factors included in trees are reported in Table 1.

A major criticism of the local and landscape conservation literature is that studies typically focus solely on measures of species richness and rarely take advantage of species rarefaction methods. We therefore used the results of the CIT of species richness to guide further analysis by comparing sample-based rarefaction curves (MaoTao estimations in EstimateS) of spider richness in partitions of high and low elevation (elevation was the most important factor). All data were analyzed in R version 2.15.0.

Results

The estimated species accumulation curve approached asymptotic species richness for this arboreal spider community, suggesting our sampling had captured a significant portion of the arboreal spider community (Fig. 1). There were 934 spiders collected in total from the sites, consisting of 109 morphospecies. Only sexually mature spiders were included in the abundance data and about 15% of the samples consisted of spiders that were not sexually mature. The composition of the canopy spider communities was comprised mainly of spiders of the families Theridiidae with 44.4% and Anyphaenidae with 20.9%. The most abundant species was Theridion nudum (Levi, 1967) of the family Theridiidae with 168 individuals followed by Wulfila inornatus (Pickard-Cambridge, 1898) with 139 individuals and Teudis geminus (Petrunkevitch, 1911) both in Anyphaenidae.

Figure 1 Estimated species accumulation across all samples.

The solid lines represent 95% confidence intervals.

Results from GLMMs

Elevation consistently had a negative relationship with total arboreal spider richness and abundance, web-building spider richness and abundance, and wandering spider richness and abundance (Table 2). Branch volume was the next most important predictor and was positively correlated with total arboreal spider abundance, web-building spider richness and abundance, and wandering spider richness. Leaf area was positively correlated with total arboreal spider abundance and web-building spider abundance.

Table 2 Outputs from the generalized linear mixed models.

	Estimatea	z value	p-value	
Total arboreal spider abundance				
Intercept	4.005 ± 0.412	9.8	<0.001	
Elevation	−0.002 ± 0.0004	−5.4	<0.001	
Leaf area	0.0623 ± 0.026	2.4	0.01644	
Branch volume	0.0014 ± 0.0004	2.9	0.00343	
Total arboreal spider richness				
Intercept	2.31 ± 0.324	7.1	<0.001	
Elevation	−0.002 ± 0.0003	−5.7	<0.001	
Leaf area	0.075 ± 0.043	1.7	0.0809	
Branch volume	0.448 ± 0.263	1.7	0.0893	
Web-building spider abundance				
Intercept	3.314 ± 0.488	6.8	<0.001	
Elevation	−0.002 ± 0.0005	−4.9	<0.001	
Leaf area	0.081 ± 0.037	2.2	0.0289	
Branch volume	0.002 ± 0.0006	2.5	0.0124	
Web-building spider richness				
Intercept	1.93 ± 0.394	4.9	<0.001	
Elevation	−0.001 ± 0.0003	−3.6	<0.001	
Branch volume	0.002 ± 0.0007	2.8	0.0045	
Wandering spider abundance				
Intercept	3.03 ± 0.52	5.8	<0.001	
Elevation	−0.002 ± 0.0005	−3.7	<0.001	
Leaf area	0.09 ± 0.033	2.9	0.0041	
Inga connections	0.13 ± 0.048	2.7	<0.001	
Wandering spider richness				
Intercept	1.35 ± 0.369	3.6	<0.001	
Elevation	−0.0008 ± 0.0003	−2.5	0.0132	
Branch volume	0.0019 ± 0.0008	2.5	0.0137	
Notes.

a Estimates on Table 2 include ± standard error.

Results from CITs

Like the GLMM, the results from the CITs suggested that elevation was the most important factor explaining richness and abundance of spiders. However, the CITs also revealed additional factors that were significant descriptors of changes in arboreal spider communities.

Overall arboreal spider abundance

At low elevations, larger branches had higher overall abundance. However, smaller branches in plots with high tree density had higher abundance of spiders than did small branches with low tree density. Finally, in those plots with low tree density, high percent forest in the surrounding landscape positively correlated with spider abundance (Fig. 2).

Figure 2 CIT total arboreal spider abundance.

The p-values are listed on each node inside of the encircled explanatory variable which responded strongest to web-building spider richness. The inner-quartile range of the data is shown in the box plot where the dark horizontal line shows the median and the whiskers show 1.5x inner-quartile range. Circles above the whisker show points that fall beyond 1.5x inner-quartile range. The number of data points (n) is shown above each box plot.

Wandering spider abundance

Similar to overall arboreal spider abundance, low elevation sites with larger branches had the highest wandering spider abundance. In the plots with smaller branches, having a higher density of trees in the plot correlated with greater wandering spider abundance (Fig. 3).

Figure 3 CIT wandering spider abundance.

The p-values are listed on each node inside of the encircled explanatory variable which responded strongest to web-building spider richness. The inner-quartile range of the data is shown in the box plot where the dark horizontal line shows the median and the whiskers show 1.5x inner-quartile range. Circles above the whisker show points that fall beyond 1.5x inner-quartile range. The number of data points (n) is shown above each box plot.

Web-building spider abundance

Sites that were at or below 623 m had significantly higher abundances of web-building spiders. At intermediate elevations larger branch size correlated with higher web-building spider abundance. Furthermore, at high elevations having at least one Inga spp. canopy touching the sampled tree canopy correlated with higher web-building spider abundances (Fig. 4).

Figure 4 CIT web-building spider abundance.

The p-values are listed on each node inside of the encircled explanatory variable which responded strongest to web-building spider richness. The inner-quartile range of the data is shown in the box plot where the dark horizontal line shows the median and the whiskers show 1.5x inner-quartile range. Circles above the whisker show points that fall beyond 1.5x inner-quartile range. The number of data points (n) is shown above each box plot.

Overall arboreal spider richness

Low elevation sites had significantly higher richness than high elevation sites. In low elevation sites large branch size was positively correlated with higher richness of overall arboreal spiders. High elevation sites with greater than 25 percent shade cover were positively correlated with higher arboreal spider richness (Fig. 5). Although the CIT reported significant differences in species richness and high and low elevation, we found no differences between the accumulation of species in sites at high or low elevations (Fig. 6).

Figure 5 CIT total arboreal spider richness.

The p-values are listed on each node inside of the encircled explanatory variable which responded strongest to web-building spider richness. The inner-quartile range of the data is shown in the box plot where the dark horizontal line shows the median and the whiskers show 1.5x inner-quartile range. Circles above the whisker show points that fall beyond 1.5x inner-quartile range. The number of data points (n) is shown above each box plot.

Figure 6 Estimated species accumulation curves for high and low elevations.

High elevation sites (black; >740 masl) and low elevation sites (white; <750 masl). The thin solid lines and dotted lines represent 95% confidence intervals for high and low elevation sites respectively.

Wandering spider richness

Sites that had more than two Inga spp. canopies touching our sampled tree were positively correlated with wandering spider species richness. In sites with two or fewer Inga spp. canopy connections, having a less heterogeneous landscape was correlated with higher wandering spider species richness (Fig. 7).

Figure 7 CIT wandering spider richness.

The p-values are listed on each node inside of the encircled explanatory variable which responded strongest to web-building spider abundance. The inner-quartile range of the data is shown in the box plot where the dark horizontal line shows the median and the whiskers show 1.5x inner-quartile range. Circles above the whisker show points that fall beyond 1.5x inner-quartile range. The number of data points (n) is shown above each box plot.

Web-building spider richness

Sites with low elevation under 810 m correlated with higher web-building spider richness. Similar richness was observed at higher elevation only with larger branch size and at least one Inga spp. canopy touching the sampled tree canopy (Fig. 8).

Figure 8 CIT web-building spider richness.

The p-values are listed on each node inside of the encircled explanatory variable which responded strongest to web-building spider richness. The inner-quartile range of the data is shown in the box plot where the dark horizontal line shows the median and the whiskers show 1.5x inner-quartile range. Circles above the whisker show points that fall beyond 1.5x inner-quartile range. The number of data points (n) is shown above each box plot.

Discussion

This is the first study, to the knowledge of the authors, to sample canopy spiders in coffee plantations. We found that different groups of spiders are affected in different ways by tree, plot and landscape scale factors. Branch size was consistently important in predicting arboreal spider abundance, and while it can be taken as a measure of sampling effort across the sites, it can also be useful when determining management practices across the coffee plantations. There is extensive trimming of shade trees across almost all of the plantations and these results suggest more robust tree canopies lead to greater overall arboreal spider abundance and richness.

With the exception of the wandering spider species richness, elevation was the strongest and most consistent predictor of arboreal spider abundance and species richness. With increasing elevation we found a decrease in both abundance and species richness. The elevation gradient in species richness has been well studied for many organisms (reviewed by Willig, Kaufman & Stevens, 2003; Hodkinson, 2005) including spiders (Otto & Svensson, 1982; Urones & Puerto, 1988; Olson, 1994; Russell-Smith & Stork, 1994; Rahbek, 1995; Bowden & Buddle, 2010; Stenchly et al., 2011). Although a number of studies have found a negative relationship between the species richness and abundance of ground dwelling spiders and elevation (Otto & Svensson, 1982; Rushton & Eyre, 1992; Chatzaki et al., 2005), others have found no effects or an increase in the abundance of certain groups (Urones & Puerto, 1988; Russell-Smith & Stork, 1994; Chatzaki et al., 2005). For example Stenchly et al. (2011) studied web building spiders in cacao plantations in Indonesia and reported a positive relationship between spider abundance and elevation. The inconsistencies in distribution pattern could be due to the variability of elevational ranges and types of spiders included in the studies, as well as the potential impact of other variables such as habitat types and landscape heterogeneity. In our study, the lack of response for wandering spider richness suggests that this group of spiders is less sensitive to elevational gradients.

We found some support for our expectation of increased spider abundance and richness with greater shade cover and tree density management. At the plot scale of 1,000 m radius, 25% shade cover significantly increased overall arboreal spider richness. The number of trees per plot had a positive effect on the overall arboreal spider abundance, and abundance of wandering spiders. Inga spp. trees tend to be the most common trees in many of the coffee plantations in the region because of their association with nitrogen fixing bacteria (Moguel & Toledo, 1999). In some of the more intensive plantations they account for around 90% of the non-crop trees. Web-building arboreal spider abundance and the richness increased with the number of Inga spp. tree connections to the focal tree.

Overall, landscape scale effects were absent or weak in our study. Similar to what Stenchly and colleagues (2011) found for web-building spiders in cacao plantations in Indonesia, we did not detect any effect of distance to forest, and this may be because the coffee agroecosystem offers habitat architecturally similar to the forest. Although never the most explanatory variable, the proportion of forest surrounding sites did have some minor impacts on spiders, it was positively correlated with the abundance of all arboreal spiders and a negatively correlated with the richness of web-builders. In the case of the overall abundance of arboreal spiders the positive correlation was found only at lower elevation site (below 920 m), lower branch volume (≤ 204 cm3) and in plots with lower tree density (≤ 165 individuals). It seems that, at lower elevations, where spiders are more abundant, and under conditions of lower vegetation density (more intensive sites) the forest acts as a source for arboreal spiders. The lower richness in web-building spiders with greater proportion of forest was observed at higher elevations (>811 m) where spider abundance and richness was low, as well as with smaller branch sizes. This suggests that at higher elevations web-building spiders are more forest specialists and don’t move much into coffee plantation. Land use heterogeneity has been shown to be important to arboreal spider communities in tropical agroforestry systems (Stenchly et al., 2011) and we detected a negative correlation between wandering spider richness with an increase in land use heterogeneity.

The surprisingly large and consistent correlation of elevation across most groups of arboreal spiders can have very important implications in light of climate change. Arboreal spiders were consistently higher in abundance and richness at lower elevations, and as the range of elevation for coffee cultivation dwindles the services provided by spiders may become more important. It is estimated that within a 2 °C change in global temperature there will be a 400 m elevational shift in suitable coffee growing range (Vermeulen et al., 2013). Our results suggest that as less coffee is grown in lower elevations, the pest control services of the more diverse and abundant spider communities will be lost for coffee production. In the foreseeable future there will be new limitations on the elevation range of coffee cultivation and along with this comes potential threat of increases pest densities. One of the most globally important pests of coffee, the coffee berry borer, thrives at the higher temperatures assured by global climate change (Jaramillo et al., 2011). An increase in berry borer abundance and decrease in spider pest control services has the potential to negatively affect yields of across the globe.

In the light of future hardship, proactive management practices can be set into motion that will promote abundance and diversity of arboreal spiders and make coffee systems more resilient to global climate change. Having more trees, greater canopy cover and greater canopy connectivity results in more abundance and richness in arboreal spider communities. Not only do these management practices increase arboreal spider diversity, but also an emphasis on high shade grown coffee can lead to more than a 10% increase in coffee production and a consistently cooler microclimate within the coffee agroecosystem (Jaramillo et al., 2013). This cooler and less variable microclimate in shade coffee leads to lower proportions of coffee berries infested by the coffee borer than on sun grown coffee plantations (Jaramillo et al., 2013).

This study demonstrates that coffee agroecosystems with more trees, greater shade cover, and greater canopy connectivity harbor greater abundance and richness in spider communities, particularly at lower elevation where spider richness and abundance tends to be higher. This result has practical management applications that suggest having shade grown coffee offers more suitable habitat for arboreal spiders due to a variety of the characteristics of the shade trees. Our results show consistently more diverse arboreal spider communities in lower elevations and this result is important in light of looming global climate change and the trends of increased management intensity. As the range of suitable elevations for coffee cultivation shrinks, less intensive management practices which promote arboreal spider diversity will be important in sustaining the viability of coffee and the livelihoods of those producing it.

We thank Pedro Perez-Lopez for assistance in the field, Guillermo Ibarra Nuñez for help with identification of spiders, and the farms for allowing us to conduct the research. Erin Catherine Sears, Fletcher Henderson, and Ferdinand LaMothe for helpful comments and ideas.

Additional Information and Declarations

Competing Interests

Author Contributions

The authors declare there are no competing interests.

Zachary Hajian-Forooshani and David J. Gonthier conceived and designed the experiments, performed the experiments, analyzed the data, contributed reagents/materials/analysis tools, wrote the paper, prepared figures and/or tables, reviewed drafts of the paper.

Linda Marín conceived and designed the experiments, performed the experiments, analyzed the data, contributed reagents/materials/analysis tools, reviewed drafts of the paper.

Aaron L. Iverson and Ivette Perfecto contributed reagents/materials/analysis tools, reviewed drafts of the paper.

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
