# Peer review of "Changes in species diversity of arboreal spiders in Mexican coffee agroecosystems: untangling the web of local and landscape influences driving diversity"

_PeerJ, doi:10.7717/peerj.623_

## Round 0.1 · original submission · Minor Revisions

The reviewers, especially reviewer two, provided exceptionally detailed and thorough recommendations for your manuscript. In looking at their comments, I do not find any core issues with your manuscript. Instead, improving the clarity of your text and addressing a number of individual issues are necessary. I think raised by the reviewers, and after addressing those issues I expect the manuscript will be acceptable for publication.

·

Basic reporting

The manuscript “Arboreal spiders in coffee agroecosystems: Untangling the
web of local and landscape influences driving diversity” addresses important issue in landscape ecology subject. Authors claim this study may be the first report on arboreal spiders associated to coffee plantations. The manuscript is well presented although some typo and grammar issues should be checked (see comments for the author). Authors present a good background on the subject. Overall the methods and analysis were properly conducted. Conclusions are supported by the data presented. The number of tables and figures was sufficient to show the results. Therefore, although minor corrections are still necessary, the manuscript is suitable for further publication.

Experimental design

No comments.

Validity of the findings

No comments.

Additional comments

Please check the following:

Page Line(s) Comment
3 20-23 The sentence “Some species response to different scales of management intensity, (Tscharntke et al. 2005, Drapela 2007, Batary 2008) therefore management at local and landscape levels can have varying impacts depending on the species.” should be rewritten. It seems authors meant that some species respond. If so, our suggestion is: “Some species respond to different scales of management intensity (Tscharntke et al. 2005, Drapela 2007, Batary 2008), therefore management at local and landscape levels can have varying... “ . Check sentence.
4 53-54 Consider chronological order for the references “Schmidt et al. 2008, Schmidt et al. 2005, Clough et al. 2005”
5 90-100 Include reference or detail some of the procedures. For instance, was the number of leaves counted or estimated? How?
8 137 (De’ath and Fabricus 2000) is missing in the literature cited
8 138-139 (Piramuthu 2008) is also missing in the literature cited
8 141 Remove period of “Table 1.” Also, rewrite the sentence. Consider replacing the expression “...to see what are...”.
8 151-152 Check grammar
9 157 Use Table 1 (capital initial) for consistency
9 171 In the reference (Hämmer, 2001; Anderson, 2001), Hämmer is not in the literature cited. Please consider alphabetical order in this case because both references are from 2001
13 250-253 The sentence “Branch size was consistently important in predicting arboreal spider abundance and can be taken as measures of sampling effort across the sampled sites, but this can be informative since both
measures can be useful when determining management practices across the coffee plantations” should be rewritten. Check grammar. Also, are the measures branch size and abundance? The overuse of “measures” should be avoided in the sentence because it may affect reader’s comprehension.
13 259-260 In the references “...Urones and puetro 1988... and ... Bowden and buddle 2010...”, use Puetro and Buddle instead (first letter capitalized). Also, the reference Bowen and Buddle is not in the literature cited.
14 284-286 Check grammar
14 289 Instead “...where spiders are more abundance...”, prefer “...where spiders are more abundant...”
15 294 Check the phrase “...and don’t more much into coffee plantation.”. Check grammar.
15 295 Remove comma
16 All lines within the item Literature cited should be entirely reviewed because some references were not cited. Check for consistency and forms too.
17 350-351 The citation “Foley, J. A., DeFries, R., Asner, G. P., Barford, C., Bonan, G., Carpenter, S. R., . . . Gibbs, H. K. (2005). Global consequences of land use. Science, 309(5734), 570-574.” is repeated.
18 376-377 Use small letter in “SOTOPINTO” for consistency.
Table 1 Define CBH

Reviewer 2 ·

Basic reporting

(i) Article structure: "Materials & Methods" instead of "Methods" (see Basic Manuscript Organization)
(ii) Minor revisons are needed to make the manuscript more fluent for reading (please see General comments)

Experimental design

"No comments"

Validity of the findings

"No comments"

Additional comments

The submitted manuscript entitled "Arboreal spiders in coffee agroecosystems: Untangling the web of local and landscape influences driving diversity" handles a compact study of shade tree dwelling spider communities in Mexican coffee plantations with respect to their spatial changes influenced by local and landscape scale management. The apparently simple but highly effective beating method in entomological and arachnological sciences can gives us comprehensive and valuable data on arthropod species richness in agroforestry systems, as the sample size with a total of 114 sampled trees is perfectly adequate for the presented study. But I think that the overall analysis of the data hasn't quite turned out as I'd hoped while reading the introduction, since this data set offers far more results than being considered by the authors so far. Furthermore, I regret to say that I found a few inconsistencies within the text as well as questionable discussion approaches, that are based in most cases on a complicated and nested language use. Likewise, text blocks are often not placed properly, which makes a fluent reading and an understanding difficult for the reader.

Nevertheless, this study revealed new insights into spiders' ecological requirements within modern coffee agroforestry systems and how their community structure might be influenced by management intensification and climate change prospectively. Hence, I recommend a future publication of these results, however with the submission of major revisions.

In the following I will list the most important things that I have noticed and that should be considered for revision:

Introduction

Line 11: Only a suggestion to make reading more fluent
"Agriculture has the potential....worldwide. Having currently about 40% of the terrestrial Earth under agricultural land-use (...) the need for a more effective management of agroecosystems considering both food production and biodiversity conservation is evident."
Line 20: This sentence does not make sense to me. Most (not only some) species show responses when changing managment intensity. But I understand the context "species show responses at different scales", wherefore this sentence should be reformulated.

Line 26: A suggestion to make reading more fluent
"Intensification of coffee often consists of an increase in chemical inputs and of a reduction in shade tree diversity, shade tree density and thus overall canopy complexity.

Line 29: "In cacao plantations, for example, creating and managing a high shade tree density can lead to an increase in the abundance of important generalist predators in cacao trees, such as web-builing spiders (Stenchly et al. 2011)."

Line 32: This paragraph that focusses on the functional role of spiders should be rewritten. Many information are given twice and should be avoided.

Line 42: I suggest to combine both sentences:
"For example, coffee-dwelling spiders....() and spiders that live on tree trunks.....().

Line 48: Strictly speaking:
Similarly, in cacao agroforestry systems, Stenchly et al. (2012) reported no impact of shade tree density on overall spider species richness or spider abundance, but a positive impact on the abundance of cacao tree canopy inhabiting web-building spiders (Stenchly et al. 2011).

Line 50: The statement of "Spiders lack of response or positive response to shade intensification" does not fit in conjunction to the previous statements, where shade tree density seems to have only a marginal effect on spiders. Or did authors meant "management intensification"?

Line 54:
"Spiders in tropical agroforestry systems and temperate arable crops appear to
respond to different factors, so understanding what makes these assemblages respond differently is important."

I do not think that spider responses on managment intensification, let`s say herbicide usage, vary significantly among those both habitat types. I just think that it is obvious that effects resulting from an high-input management in coffee or cacao are masked by the overall high habitat heterogeneity that agroforestry systems offer. In other words, spiders in homogenaous habitats such as wheat fields, will be automatically more affected by environmental changes than spiders of agroforests. Spider species that can be found in common crop fields in temperate zone, already live on the edge of their well-being and every addition to this system (such as diverse field margins) will have a prompt posititve impact. This is why the right interpretation of results coming from studies focussing on agroforestry systems like yours, are much more tricky.

Line 58: Sentences "Work by Stenchly...agroecosystems." should be deleted.

Line 61 - 83: This paragraph must be shortened and revised carefully with respect to language and content because stating clearly the research objectives and working hypotheses are crucial for the importance of the manuscript. Unfortunately, both are not written very well.

One suggestion how to write-up this paragraph:

..."Hence, this study was carried out to gain insights which factors are important drivers of arboreal spider diversity of shade trees in coffee agroforestry systems while considering tree characteristics, plot management intensity and landscape features. We hypothesized that a higher canopy connectedness as a result of high shade tree density effects an increase in spider species richness and abundance on tree scale due to facilitating habitat access for arboreal spiders. It is further hypthesized that a more dense shade tree layer promotes high spider abundance on plot scale, as this positive relationship could be already reported for tropical spiders in cacao agroforestry systems (Korinus 2007). At Landscape scale, distance to forest is expected to have no impact on spider communities of shade trees in coffee plantations. However, it is hypothesized that the degree of forest cover in a 1000 m radius will have a negative impact on spider richness and abundance based on the assumption that spiders will remain within forest patches and not move to the coffee patches because unmanaged land tend to harbor highly diverse communities (Batáry 2012). Furthermore, at landscape scale elevation has been an effective predictor of spider communities in the tropics (Russell-Smith & Stork, 1994, Stenchly et al. 2011), and we assumed a decrease in abundance and species richness with increasing elevation.

Introduction - in general:

Authors should also be more unique with respect to the use of the terms "scale" and "level". It is advisable to use only one term and this througout the whole mansucript. Otherwise the raeders get the impression that this mansucript was not sufficiently thought-out by the authors.

Until line 114 it is not clear to the reader that the actual protagonists of this study are the shade trees and not the coffee plant itself, in this case Inga spec.. This should be stated clearly already within the Introduction by giving the link between the value of shade trees in coffee agroforestry systems as additional or main habitat for spiders or as valuable climate stabilizers and their resulting impact on overall biodiversity.


Material and Methods

Line 86: "...ranged in elevation between 695-1273 m asl." - This values differs from that listed in Table 1 having a minumim elevation of 595 m. Requires revision, either wihtin text or table.

Line 87: I would include: "...by coffee agriculture (94% land cover)..."

Line 89: Authors wrote that the selected nine coffee planations varied by management intensity. However, management intensity also includes the use or non-use of herbicides, pesticides and fungicides. Because Inga spec. is the dominant shade tree species, I assume that plots were located in modern coffee agroforestry systems with regular chemical application. If so, authors should ensure that all plots were treated to the same degree and to the same time and if not, than chemical input should be considered as possible factor that shapes spider abundance/richness. Chemical application can have tremendous impact on spider abundance leading to biased results, particulary in temporal restricted studies such as one time beating.

Line 92: One tree scale parameter constitutes the "number of tree canopies touching the sampled trees". I think it would be much more descriptive if the area of canopy overlap would be taken as factor. If authors also measured the overlap area, I suggest to use this instead of the number.

Line 99: "We measured shade cover at the center, 5m and 10m away in each cardinal direction and used the average of these measurements." I do not understand the term "shade cover". I assume that authors meant "canopy cover"? And if so, which method was used to measure the canopy cover (e.g. densiometer)?

Line 101 - 113: This section confuses me. Is shade intensity level the same as average percent shade cover? I suggest to rewrite this paragraph and authors should be conistent with factor terms to make it easier for the reader to follow the experimental set-up.

All used statistical analysis working fine for itself, however, authors should be careful by chosing the one apropriate statitsic that really helps to answer their research question. Particularly with regard to data analyses using GLMM and CIT. These are completely different approaches what makes data interpretation difficult. The basis for using GLMM is to assume a linaer relationship between response variable and factors. And a huge advantage of GLMMs is the conisderation of pseudo-replication, while putting "site" as random factor. And this is something that CITs do not consider, wherfore the outcomes of this tool might be biased. Honestly, I do not know how CIT analyses can handle multicollinearity and unfortunately I could not find the reference "Piramuthu 2008" within the reference list. However, after some research I found an article of Tagliamonte & Baayen, 2012 stating that "The conditional permutation variable importance implemented in the cforest function of the party package correctly reports spurious predictors to have a very low variable importance." Nervertheless, I like both analysis approaches wherefore both should presented and dicussed.

The Material and method section requires also a revision of the language. I also suggest to delete the first paragraph because some information are given twice. Authors should start directly with GLMM.


Results
The totalling of the n-values showed in Figure 1 - Figure 4, revealed a total n of 109 sampled trees. However, by taking 38 sites with 3 sample trees per site into account, I come to a total n of 114 sampled trees. Did I miss here something or were some trees taken out of the analyses and I red over it?

Line 176: Authors wrote "There were 934 spiders collected in total..." Does this number only consider adult spiders or the total number including also immature spiders? If this is the total (adults+immature) I wonder at the really low individual number. Considering that in total 114 (109) trees were sampled, I calculated an average spider abundance of 8 per tree or 4 spiders per branch. This is not much. How can this overall low spider abundance be explained?

Line 179: this should be called "spiders of the families" not "in the families". See also Line 181 and 183.

Line 185: Sentences like "Outputs from all GLMM can ..." should be avoided. Better: "Elevation consistently had a negative effect across total arboreal spider richness and abundance, web-building spider richness and abundance, and wandering spider richness and abundance (Table 2)."

Line 189: Better: "Leaf area was positively correlated with total arboreal spider abundance and web-building spider abundance."

Line 193: Better: "However, the CITs also revealed additional factors that were significant descriptors of changes in arboreal spider communities."

The next sections (results of CITs) should be summarized in a more uncomplicated way, wherfore I suggest to present the results, firstly without subheadings and secondly I suggest that author should present the results of richness and abundance in a more compact way.

Line 226: How authors defined "low" and "high" elevation? Figure 2, for example, differentiates spider communities in above and below 920 m a.s.l. - However, Figure 5 differentiates communities above and below 774 m a.s.l. And compositional analyses considered 740 m a.s.l. as threshold. Such inconsistency should by avoided.
Furthermore, the NMDS shows clearly that the number of sampled tress in higher elevations is more than twice as high as in lower elevation. Hence a comparision of high vs. low communities is not advisable. Appropriate data analysis should be planned in advance, before the experimental set up. Hence, I suggest to not include composition analysis. However, it would be intersting to see the results of a Manteltest.

Figure 2: "abundance" is missing at y-axis

Discussion

Maybe I missed it, but for readers it would be very informative if authors could show how environmental paramters are correlated to each other (multicollinearity). Because I assume that shade cover, number of trees per plot and Inga spp. tree connection are closely related (Line 271).

Also the Discussion section requires a revsion concerning language, such as "..and this is likely because coffee agroecosystems offer a habitat that is architecturally similar to adjacent forests." (Line 283) or "...where spiders are more abundant, and under conditions of lower vegetation density (more intensive sites), the adjacent forests can act as a habitat source for arboreal spiders.."(Line 290).

Line 303: When talking about community changes along an altitudinal gradient, we have to consider the actual parameters that are correlated with elevation, such as temperature and habitat architucture. Particularly temperature range and minimum temperature seems to be important factors for spider distribution and other arthropods such as insects. Hence, when talking about climate change and its impact on distribution pattern of pest insects in agricultural systems, we also have to look at the future pattern of their predators. Will they adapt to higher elevations to the same degree as their prey? Hence, I do not think that the biocontrol service provided by spider will get lost, as stated by the authors in Line 305, du to the overall shift of arthropod abundances/communities. It is more likely that intensification processes will lead to the impoverishment of preadator communities.

Further minor revisons are needed to make the Discussion section more fluent for reading. The syntax is often very confusing and formulations such as "...don`t more much into coffee planatation" (Line 294) should be avoided. Authors could let end the sentence after "... more web-building spider species seems to be forest specialists."

Some minor issues:

Title - I suggest to change the title to: Changes in species diversity of aboreal spider communities in Mexican coffee agroecosystems: Untangling the web of local and landscape scale effects

Citations - I noticed that authors missed to write "et al." (e.g.: Richert 1984 or Schmidt 2005) However, if more than three authors were involoved in one article, it must be stated via using "et al." These corrections have to be made throughout the entire manuscript.

Please correct to: Russell-Smith, A., & Stork, N. (1994). Abundance and diversity of spiders from the canopy of tropical rainforests with particular reference to Sulawesi, Indonesia. Journal of Tropical Ecology, 10(4), 545-558.

---

## Round 0.2 · accepted · Accept

Thanks for your thorough revision, and detailed attention to reviewers suggestions. I think it's a strong paper, and I'm happy you chose PeerJ.